# The Grass Ceiling: Hidden Educational Barriers in Rural England

Luke Graham 

School of Education, University of Exeter, St Luke's Campus, Exeter EX2 4TH, UK; l.graham@exeter.ac.uk

**Abstract:** Rurality is rarely integrated into analyses of educational inequalities and this article presents an alternative perspective on rural–urban attainment and highlights the impact of rurality on educational outcomes. The traditional narrative of urban–rural educational disadvantage is that urban pupils do less well in the English exam system. Decontextualised data across different English exam performance measures demonstrate how rural pupils outperform their urban counterparts. Socioeconomic disadvantage (SED) has the most significant impact on attainment and this analysis explores the rural–urban attainment gap through this SED lens. An analysis of the Department for Education (DfE) data explores possible factors that might explain the emerging rural educational gap and presents an argument that rurality is another limiting factor that intersects with SED. This article demonstrates how rural underachievement in England has been hidden by the relative sizes and SED distribution of rural and urban populations.

**Keywords:** rural; education; attainment; England; disadvantage; exam outcome; GCSE

## 1. Introduction

Educational performance is an important factor in shaping adult opportunities, including employment, income, and wealth [1]. However, there are marked socio-economic gaps in performance that emerge early in life and widen throughout childhood [2] as a consequence of pupils' experiences during school [3,4]. Pupils from disadvantaged backgrounds perform less well in statuary examinations and are less likely to participate in Higher Education (HE) [3]. There are also marked disparities across different socioeconomic groups in HE completion [5].

The English socioeconomic disadvantage (SED) gaps established in the late 1980s [6] have widened by 30–40% between the 1980s and 2000s [7]. These socioeconomic inequalities in attainment and participation in school and HE have proved to be 'stubbornly durable' [6] and the SED attainment gap is widely recognised across the world [8–17].

SED (proxied for example by receipt of Free School Meals (FSM) or parental educational attainment) is intersected by other dimensions of educational performance, such as gender. These factors overlap and amplify [18–20] and by the time pupils reach post-18 education, these gaps are even more significant [21,22]. Rurality is rarely integrated into these analyses of educational inequalities [23–26] but recent changes to DfE data include rurality measures and this exposes some previously hidden barriers to educational participation [27–30].

*The Impact of Disadvantage*

There are many persisting stratified social patterns in educational outcomes [31], or *gaps* in education, and particular focus is given to the well-recognised gaps. The achievement (or attainment) gaps are an example of unfair distribution [32] and are the differences in achievement between different groups of pupils that persist as an intergenerational reproduction of inequality [33].

Irrespective of how educational attainment outcomes are measured, particular groups of pupils do less well across all phases of schooling, including post-16, and post-18 education. There are several attainment gaps but the most persistent are the manifestation of the issues of social (in) justice in education. The gap in achievement between the most and least socio-economically disadvantaged (SED) [34] is the most significant [35] but there are other well-recognised participation patterns of English educational outcomes associated with ethnicity [36,37], gender [38], and disability [39–41]. In England, those pupils most commonly on the wrong side of the gap tend to be SED, white, and male but for science, maths and engineering they tend to be SED, white, and female.

Gaps gain political weight because they make patterns in measured outcomes visible. Without the visibility, the gaps would not be recognised, and there would be no policy intervention. The narrow measurement of gaps focusses the political and subsequent financial attention to address the inherent injustices in education more broadly. These gaps clearly existed before they were visible; SED pupils underperformed in the system before we collected and presented data in a way that allowed the pupils to be separated and categorised. These differences were made visible only when the data revealed them. It is only when the narrow outcomes are presented in a way that identifies the different pupil characteristics that a gap is seen through a quantitative lens. To be recognised, and hopefully treated, a gap needs to first be visible.

Despite the inconsistent definition of 'disadvantage' identified below, both within government agencies and between different territories, there is a persistent and stubborn pattern. SED pupils perform less well in comparative outcomes.

Rural education is an emergent factor in the United Kingdom (UK) education ecosystem [42] and one that has increasing resonance. The House of Lords (Chapter 1, para. 27) [30] has called for a strategy to support the rural economy, partially due to the uncertainty of the UK's departure from the European Union. They identified the issues of rural access to education as a key barrier to rural economic development.

This paper aims to answer the question: is rurality in England an under-recognized educational gap?

## 2. Methods

This analysis draws on UK national data sets from different government departments using data from the 2018/19 academic year to minimise any impact of COVID-19. Table 1 summarises the data sets used.

**Table 1.** Sources of data analysed.

| Department | Document | Location |
|---|---|---|
| National Statistics/DfE | GCSE and equivalent results: 2017 to 2018 [43]. SFR06 | https://www.gov.uk/government/statistics/gcse-and-equivalent-results-2017-to-2018-provisional accessed on 15 January 2022. |
| DfE | National Pupil Database—GCSE performance by local authority [44] | https://www.find-npd-data.education.gov.uk accessed on 15 January 2022. |

These accredited official DfE statistics are reviewed by the Office for Statistics Regulation (OSR) and published as a Statistical First Release (SFR). It uses Department for Education (DfE) data on exam performance. This release uses an English deprivation measure called IDACI. The 'Income Deprivation Affecting Children Indices' (IDACI) is a composite of seven Indices of Deprivation at a small, local area level across England. It is produced by the Ministry of Housing, Communities, and Local Government (MHCLG), used by DEFRA as well as for some DfE data and ranks the relative levels of deprivation in the 32482 areas in England [45]. In much of the data published by the DfE, areas are organized in IDACI deciles, with 0–10 being the most deprived and 90–100 the least deprived.

In England, most pupils take national exams (GCSEs) at the end of High School at age 16 [46]. These data are collected nationally and made available in a collated form each year. The SFR (06_A4) provides a breakdown using the IDACI of rural and urban pupils and provides averaged GCSE outcomes and the total number of pupils in each IDACI decile. This includes all pupils from state-funded schools (including academies and other state-funded institutions). It presents the achievements at GCSE and equivalent for pupils in the final academic year of High School (age 16) by IDACI decile (based on 2010 IDACI scores) and degree of rurality of pupil residence.

### 2.1. Defining Disadvantage in England

The key terms disadvantage and rural are central to this study, and neither are consistently or clearly defined in the literature, nor within or between government departments [47].

As with any comparative research, the context in which these comparisons are made is crucial. There are additional limitations in this study in the ways that key terms are defined. It is recognised that disadvantage is a term that is deployed differently with different meanings in different contexts, and so too is the term rural.

While disadvantage has the most significant impact on attainment [48], there is an ongoing debate over the identification of disadvantage and there is no universally adopted definition [49]. The term alphabet soup, first used to describe school leadership [50], also describes the different ways SED is identified across different institutions in England. Table 2 outlines how disadvantage in pupils and students are recorded and identified by different institutions.

**Table 2.** Terminology used to describe disadvantage and underrepresentation.

| Measure | Institution Level | Description |
|---|---|---|
| Disadvantaged | Department for Education (DfE) [51,52] | Pupils who attract PP or SPP funding (see below). |
| Ever 6 | School | Pupils who have received FSM at any point in the last 6 years. |
| Free School Meals (FSM) | School | Payment made to the school to cover the cost of school lunches for pupils who live in a household in receipt of specified benefits. |
| Income Deprivation Affecting Children Index (IDACI) | Department for Environment, Food and Rural Affairs (DEFRA) | A composite of seven Indices of Deprivation at a small, local area level (Lower-layer Super Output Areas) across England. |
| Pupil Premium (PP) | School and Local Authority (LADs) | Paid to schools on a per-pupil basis for children who are Ever6, are looked after, in care, or who left care through adoption or another formal route. The PP allowance was GBP 1320 at primary school and GBP 935 at secondary (2018/19) [53]. |
| Service Pupil Premium (SPP) | School and Local Authority (LADs) | Introduced in April 2011 by the DfE in recognition of the specific challenges children from service families face. Schools in England with children of service families from reception to year 11 can receive GBP 310 per child SPP funding. |
| Tracking Underrepresentation by Area (TUNDRA) | Office for Students (OFS) | Area-based participation measure, which uses individualised and POLAR4 data. It tracks individuals from High School to Higher Education and replaced POLAR4. |

Terms such as 'persistently disadvantaged' are intermittently but inconsistently used to differentiate between Ever 6 pupils and those eligible for Free School Meals for at least 80 per cent of their time in school [54]. This attempts to add gradations to what is otherwise a binary SED measure (FSM and not FSM).

There is no universally adopted definition of SED [37] and in most analyses derived from DfE data in England, pupils are defined as disadvantaged if they are in receipt of PP [53]. However, this analysis draws on the DfE data that deploy IDACI as the disadvantage measure and includes a rural–urban classification, as defined below. This allows for detailed analysis of the links between rural outcomes controlling for deprivation.

*2.2. Defining Rurality in England*

The Rural–Urban Classification (RUC) of areas are based on the RUC2011 classification of output areas released in August 2013. Census output areas forming settlements with populations of less than 10,000 are defined as rural. The RUC2011 was the Rural–Urban Classification in place in the academic year 2018/19 [34] and is used as the rurality measure in this analysis. Despite the limitations of the RUC2011 classification outlined below [35,36], it is the favoured system of rural categorization for UK government data.

The rural–urban classification (RUC2011) additions to the DfE GCSE data [55] allow for analysis of rural–urban differences in GCSE outcomes [56] that were not previously possible and may indicate an increased political appetite to engage in conceptualisations of rurality [43]. Recent government reports [57–59] raise the profile of rurality in educational outcomes. But these data present binary notions of urban and rural by population size (as in the 2020 GCSE data [60]) and do little to recognise the conceptual and practical complexity of identifying rurality.

Despite the limitations of the rural–urban categorisation mentioned, the additional rurality data in the GCSE data [43] allows for analysis of the relationships between rurality and outcomes. This offers an opportunity to explore the patterns of rural pupils' participation in the English education system and the influence of the education system on rural pupils; however, as outlined below, de-contextualised analysis of this data has hidden some of the key patterns within it.

The data analysed uses the UK government's preferred system, the Rural–Urban Category, based on the 2011 census (RUC2011) [61]; however, the RUC2011 is problematic in two main ways.

Firstly, the RUC2011 uses a population-based rural–urban system and puts settlements of more than 10,000 in an urban category and the remaining in rural. This means more rural regions with towns with a population of 9000 will be rural, and a neighbouring town with a population of 12,000 will be urban. Urban towns in these areas are likely to share more in common with rural towns in the same area than with other urban centres classified in the same category, where the social and cultural landscape will be very different.

Secondly, rurality, as currently defined in the RUC2011 includes two different countrysides; a more affluent, less sparse, and more accessible one, and a less populous and isolated, sparse countryside [62]. Where measures of rural performance are based on the mean performance, they will be dominated by the more populous, richer, rural populations on the outskirts of towns and cities. Due to the relatively small population, the sparse, geographically isolated, remote rural areas do not make much of an impact on the national rural performance data. These two groups of rural pupils are not a cohesive whole that needs the same support for improved participation. As a result, this remote, isolated [28], rural disadvantage has remained hidden [63]. Consequently, the current rural measure available is not helpful in identifying (or subsequently closing) the rural gaps. The binary rural–urban categories (RUC2011) are not the most useful predictor of outcomes.

Within the RUC2011, there are sub-categories of rural and urban that may provide a more nuanced scale, as shown in Table 3. However, while the DfE holds this level of data, it does not make it available with the associated GCSE outcomes data for external analysis at a suitably nuanced level, so it can only be used in some of the analyses provided.

**Table 3.** RUC2011 full set of rural classifications [61].

| Rural Classification | Classification Descriptor |
|---|---|
| A1 | Urban: Major conurbation |
| B1 | Urban: Minor conurbation |
| C1 | Urban: City and town |
| C2 | City and town in a sparse setting |
| D1 | Rural: Town and fringe |
| D2 | Rural: Town and fringe in a sparse setting |
| E1 | Rural: Village |
| E2 | Rural: Village in a sparse setting |
| F1 | Rural: Hamlets and isolated dwellings |
| F2 | Rural: Hamlets and isolated dwellings in a sparse setting |

*2.3. Rural Educational Performance in England*

Urban areas have traditionally been home to larger numbers of SED and underperforming pupils. Targeted financial support in these areas benefits a larger number of pupils than could be supported in a similarly sized rural area [37]. Poverty and deprivation tend to be more geographically dispersed in rural areas than in urban areas, which can make disadvantage harder to identify [38]. Additionally, a rural area can appear to be relatively affluent and yet still include rural households in poverty [39].

Historically, when data on the rural–urban gap were presented, it was to shine a light on urban disadvantage and underperformance [27,37,40,41]. Explanations for this rural–urban gap included suggestions that rural students are more insulated from the effects of poverty, through smaller class sizes, or receiving more educational support at home [42]. With this focus on urban poverty, UK government strategies focussed on raising standards in urban schools [43,44]. Towards the end of the 1990s, the issues in educating the most disadvantaged were identified [37,45] and funding was increased to urban schools [41,44]. The outcomes of urban pupils improved over the next 12 years.

There are different measures of GCSE attainment, outcomes or performance but where possible the traditional '5 or more GCSE A* to C grades including English and mathematics' (5EM) will be presented as the measure of GCSE performance. This is the measure that schools use as the key accountability and performance measure [46]. It is also the standard minimum portfolio of qualifications needed for many pupils to enter post-16 education [47]. Other studies use the '5A* to C at GCSE' measure (5AC), where any 5 GCSE results count. As such, the 5EM bar is higher than the 5AC bar. The English Baccalaureate sets the bar higher than the 5EM as it adds additional GCSE subject requirements onto the achievement measure and hence lowers the overall percentage. While rurality is inconsistently defined across different jurisdictions [64–66], there are emergent patterns of rural–urban attainment.

Historically, the UK and Belgium have stood out as countries where rural pupils have outperformed urban pupils. While comparisons between jurisdictions are often made [67], rural research is commonly not transferable from one country to another. Even within countries, it is often not transferable from one province to another [68], and within the UK rurality is defined differently from one policy-making department to another [47].

Internationally the picture is different, and in most European and OECD countries, urban pupils outperform the rural in formal qualifications [67,69]. However, when we compare patterns across territories, we need to be mindful that national definitions of urban and rural areas differ significantly from one place to another [70], are not suitable for international comparison [65], and can even be conflicting [66]. Many countries use a minimum population size to define an urban area but that size can be as low as 200, as in Denmark [71], 1000 in Canada, 5000 in India, 50,000 in Japan, or even 100,000 in China [72] (p. 81). Some countries designate urban areas by administrative decision or through employment or provision of infrastructure and services [70].

There is still no communally accepted measure of rurality [49]. Studies at different times have explored different elements of the rural–urban spectrum. Equally, different institutions in different territories use different measures, for example, the European Union uses population density (people per square kilometre) to identify a rural area [73], rather than the threshold settlement size more commonly used in the UK [74]. Other measures used within the UK include measuring scientific or habitat features, and as of 2007, there were over 30 definitions of 'rural' used by different government departments in the UK [47], and settlements with between 3500 and 10,000 people are treated differently across these classification systems.

Additionally, methodologies and definitions used to designate rurality do not remain constant. In 2001, Australia changed the definition of 'rural' from a population of 1000 to 10,000. The United States moved from a place-based to a density-based measure in 2000. In England, the current RUC was only introduced in 2004 (based on the 2001 census) and then updated in 2011 (RUC2011). For these reasons, conclusions and observations from one context are only partially comparable across national, regional, and temporal boundaries.

## 3. Results

On averaged (decontextualized) national performance tables, rural pupils in England achieve better GCSE outcomes than urban pupils [67], equivalent to 7 to 10 percentage points at 5 + A* to C measure (5AC) [75]. DfE data for the 2017/18 academic year [76] cohort showed that when they left school, 69.5 per cent of pupils living in rural areas left school with at least English and Maths GCSEs at A* to C grade or equivalent, compared to 63.5% for urban (overall national average 64.5%) [76].

Table 4 summarises the DfE data and sheds light on how this traditional rural–urban narrative emerged. It shows the percentage of pupils from different areas who met the 5EM threshold.

**Table 4.** Urban and rural GCSE performance from DfE National Pupil Database 2018.

| Degree of Rurality (RUC 2011) | Percentage of Pupils Achieving 5 + A*-C Grades inc. English and Mathematics GCSEs (5EM) | Entering the English Baccalaureate | Achieving the English Baccalaureate |
|---|---|---|---|
| England | 53.8 | 38.7 | 24.3 |
| Urban: Major conurbation | 57.5 | 40.6 | 25.4 |
| Urban: Minor conurbation | 51.7 | 31.9 | 19.1 |
| Urban: City and town | 54.8 | 36.3 | 22.3 |
| Rural: Town and fringe | 59.4 | 40.9 | 25.9 |
| Rural: Village | 64.0 | 45.9 | 30.6 |
| Rural: Hamlet and isolated | 65.4 | 47.3 | 31.8 |

The national attainment data in England suggest that even after the substantial investment in urban schools, rural pupils are still more likely to achieve better GCSE outcomes, however it is measured. This observation is used as evidence of urban underperformance both nationally [77,78] and internationally [67]. The underperformance of urban pupils, shown in Table 4, is echoed across all collated measures of GCSE performance released by the government (for example English Baccalaureate participation and achievement shown in Table 4).

These patterns persist over time. Figure 1 is constructed from the DfE data 'Table A2' in the Statistical First Release data [43] and shows achievement at GCSE (or equivalent) for pupils at the end of High School (age 16) by the degree of rurality of pupil residence, broken into the four available RUC2011 categories from 2009 to 2014.

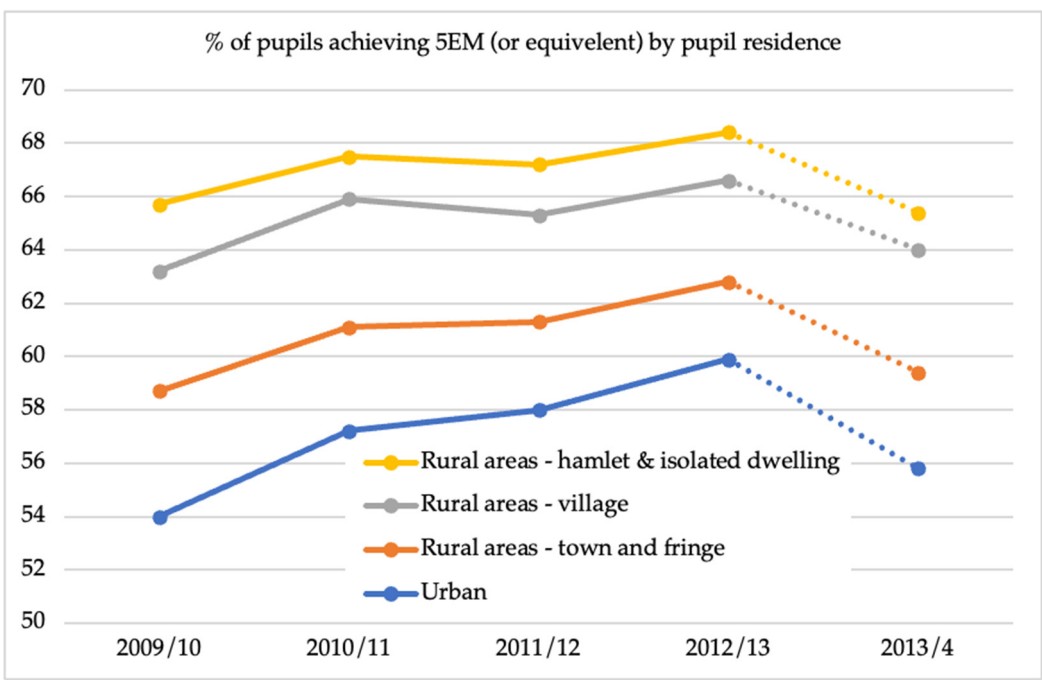

**Figure 1.** The 5EM achievement at age 16 by pupil residential area (state-funded schools in England 2009–2014). The GCSE criteria changed in 2013/4 so the scores are not directly comparable to previous years, this is indicated by a broken line in Figure 1.

There are variations in the 5EM percentage between 2009 and 2014 and an overall decrease between 2013 and 2014 due to the GCSE exam changes. However, the relative 5EM gap between rural and urban pupils has broadly endured. Irrespective of the GCSE performance measurement used or the date of analysis, rural pupils appear to do better than urban pupils. Figure 1 shows that the highest performing pupils come from rural hamlets, and they tend to be 2 percentage points (p. p.) higher than rural village pupils on the 5EM measure. In turn, rural village pupils are 4 p.p. higher than town and fringe, who are 4 p.p. higher than urban.

*3.1. Analysis through the Lens of SED*

The Income Deprivation Affecting Children Index (IDACI) measures the proportion of all children aged 0 to 15 and arranges them in order of the income of their families. The more deprived an area, the higher the IDACI score. The index is presented by the Office of the Deputy Prime Minister [79] and the data are accessible from the National Statistics as a searchable government portal. These DfE SFR data [80] include IDACI within the GCSE outcomes. Figure 2 shows the relationship between IDACI and GCSE outcomes at the 5EM level for all Local Authority Districts (LADs). Each LAD is represented by a point on Figure 2. The most deprived LADs are on the right of the graph, and the lowest attaining are at the bottom.

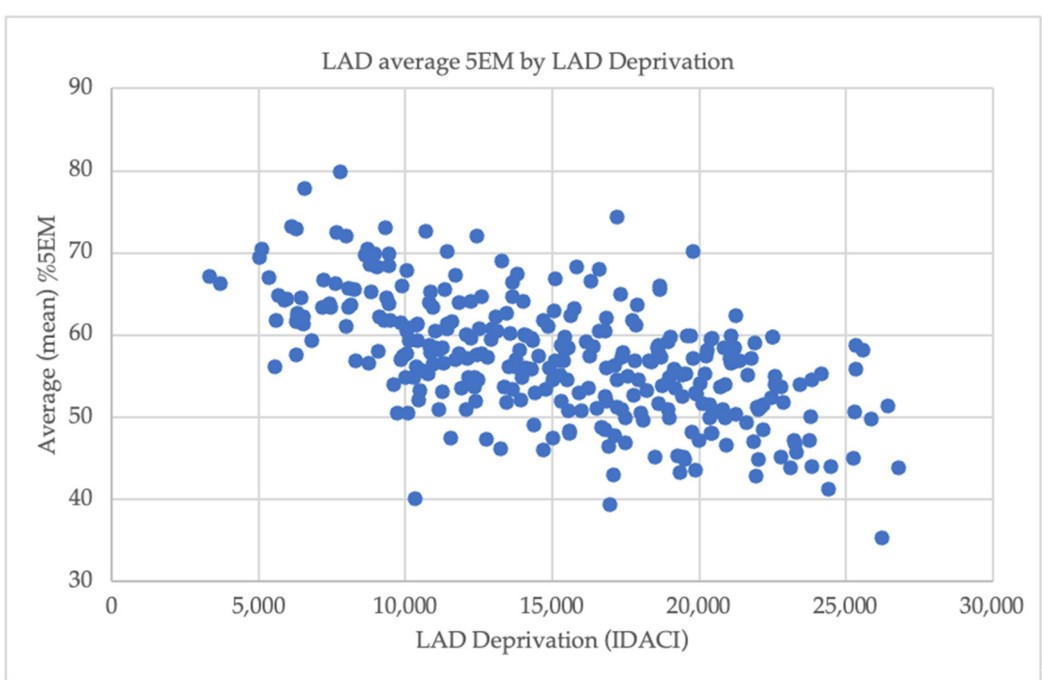

**Figure 2.** LADs analysed using IDACI deprivation and GCSE outcomes in England for 2018 data.

As expected, on average, pupils from the most affluent LADs achieve better average GCSE outcomes (Figure 2). There is a highly significant correlation between deprivation scores and average GCSE outcomes of the LADs. This is significant at the 0.01 level for both the Income Deprivation Affecting Children Index (IDACI) and the Index of Multiple Deprivation (IMD).

### 3.2. Comparing Rural and Urban Populations

As noted above, the national performance of pupils from each of these rural–urban classifications has also remained broadly constant, with rural pupils consistently 10 p.p. higher on the 5EM measure; however, the national picture obscures important issues. In part, this has to do with the relative number of urban pupils compared with the number of rural pupils. Between 2009 and 2014 (for which there is verified data), the number of pupils in the different rural classifications entering GCSE exams has remained broadly stable, as shown in Figure 3.

Around 82% of pupils taking GCSE exams in any one year are urban and 18% are rural, with half of these rural pupils living in towns and fringes [81]. The national average (mean) 5EM at GCSE will be heavily weighted by urban pupils, as they represent such a large proportion of the cohort. Secondly, the ecological fallacy [82] assumes that what is true for the average urban or rural pupil is true for individuals within that population.

Figure 4 shows the IDACI distribution for rural and urban pupils presented in 10 deciles (0–10 being the most disadvantaged).

The distribution in Figure 4 is not the same for urban and rural communities. Relatively more urban pupils come from more deprived homes when compared to rural pupils. A calculated mean IDACI for rural pupils is higher than the mean for urban pupils because of the distribution shown in Figure 4. Fewer people in rural areas live in the most deprived parts of the country, as these tend to be in urban areas [83].

It suggests that the headline finding in Table 4 and Figure 1 (that rural pupils outperform urban pupils in GCSE 5EM) may be an artefact of SED, not a factor of rurality. This difference in the relative average deprivation between rural and urban areas explains the observed difference in the GCSE performance shown in Table 4. On average, rural pupils are more affluent than urban pupils. On average, more affluent pupils do better. The ecological fallacy of comparing average pupils across two differently distributed populations

has diverted attention away from rural educational disadvantage and even introduced higher barriers for rural pupils [84].

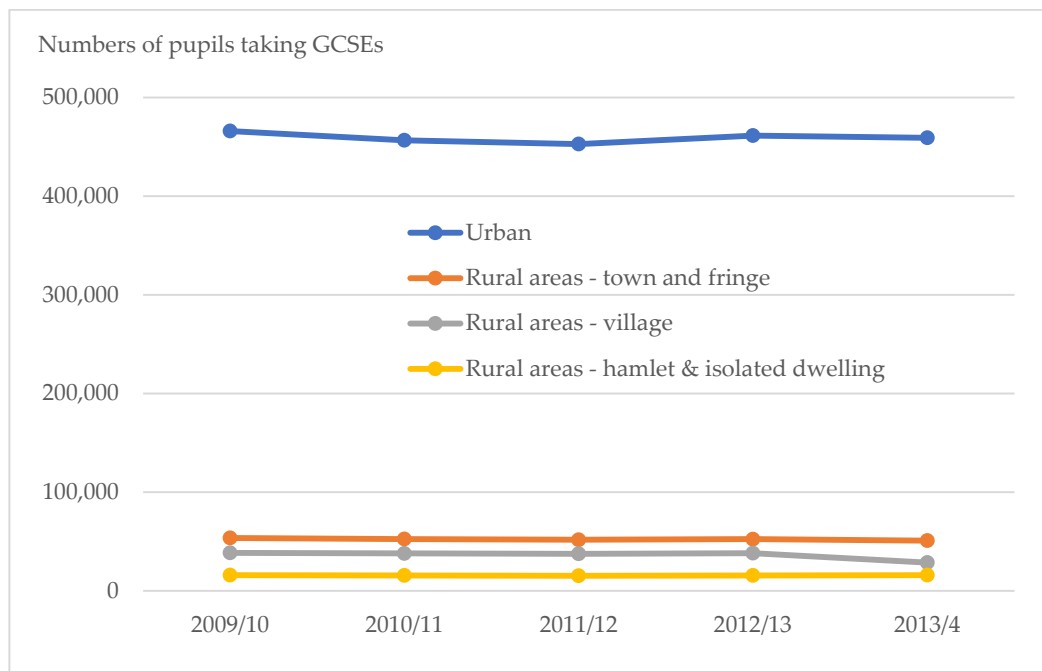

**Figure 3.** Number of pupils taking GCSEs by rurality from 2009 to 2014 data from SFR [43].

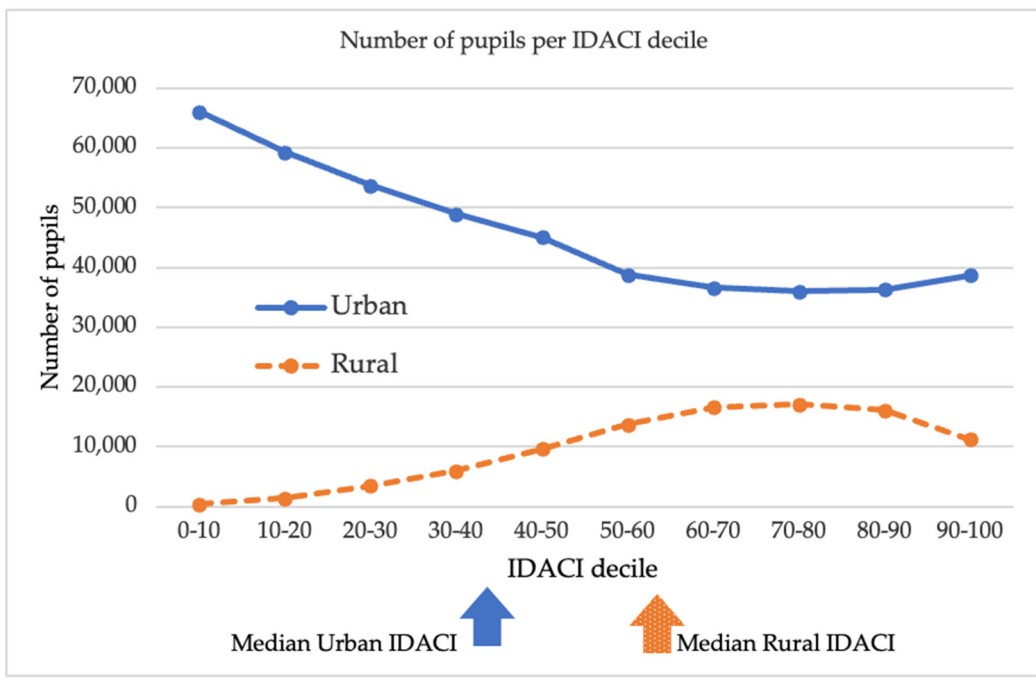

**Figure 4.** IDACI population (rural and urban) from DfE data, statistical first release 2015.

Figure 5 shows the same IDACI deciles, with the 5EM performance separated into rural and urban pupils and describes the patterns of GSCE performance through the lens of SED.

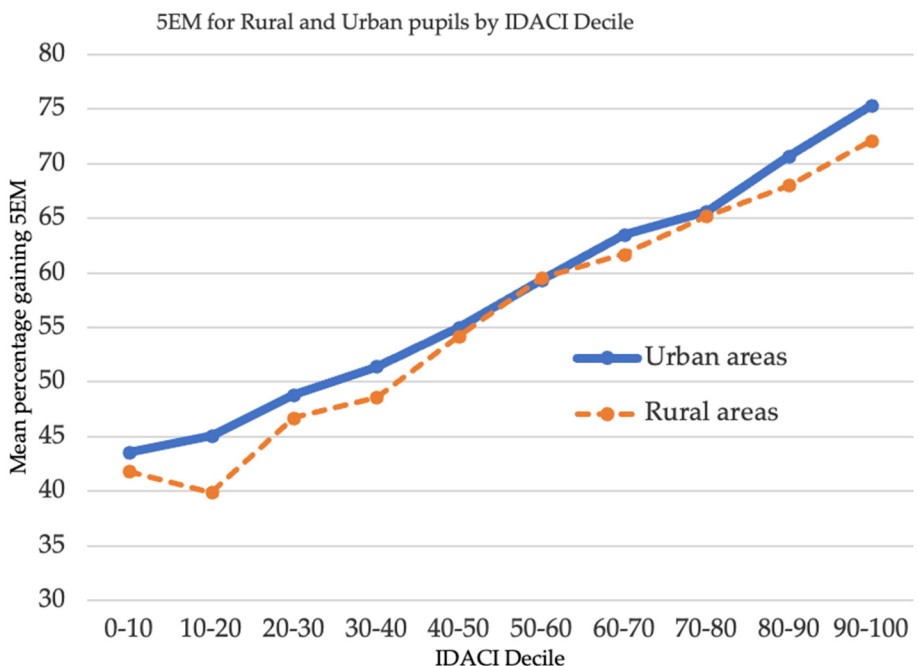

**Figure 5.** A comparative 5EM mean for rural and urban pupils at different IDACI deciles.

Far fewer of the pupils from the lowest IDACI decile (0–10) achieve 5EM. But within this, we can see how for each decile rural and urban pupils perform. The Statistical First Release that holds these data does not provide sub-classifications for urban and rural (there is no sparseness or type of rural or urban as shown in Table 3 in these data).

It is important to note that the data points in Figure 5 do not represent equal numbers of pupils. There are fewer lower IDACI rural pupils than higher IDACI rural pupils. IDACI is a population-derived decile and, therefore, includes both rural and urban pupils in its calculation. Figure 4 demonstrates how the median IDACI is very different for the rural and urban populations.

In all IDACI deciles, rural pupils **underperform** their urban counterparts on average at the 5EM level. This difference is about 2 p.p. on average. The differences are especially marked in the extremes of deprivation, where rural pupils are 4–8 p.p. lower than the equivalent urban decile. Taking account of relative deprivation removes the apparent underperformance of urban pupils. Due to the small number of rural pupils in the total school population, the mean GCSE 5EM score for the national data is highly influenced by the relatively high numbers of more affluent rural pupils. Those SED rural pupils are hidden in the population.

## 4. Conclusions

The average GCSE outcome data showing rural pupils outperforming urban pupils in Table 4 and Figure 1 are a consequence of the unequal distribution of advantage shown in Figure 4. It is a consequence of rural pupils generally being from wealthier homes. It is not because rurality improves educational outcomes. On average, urban pupils are more disadvantaged, and disadvantaged pupils do less well in education on average, therefore, average urban pupils do less well than average rural pupils. When we compare the average urban 5EM to the average rural 5EM (Table 4, Figure 1), we are not comparing like-for-like in terms of relative SED. In other words, it is the lower SED in urban areas that lowers the average GCSE outcome, not urbanness.

In England, there is an association between population size and pupil attainment. Average attainment in population-dense urban places is low. Again, this is not because urban environments disadvantage pupils but because the most disadvantaged pupils with low average attainments attend the most urbanised schools [85]. In fact, urban

schools tend to provide better outcomes for disadvantaged pupils [86], perhaps through increased competition or cooperation with other local institutions [85] or an uplift in funding [77,87,88].

The narrative that rural pupils outperform their urban peers that emerged in the early 2000s still lingers [27,57,67,89]; however, it is poverty that lowers outcomes, not urbanness [90], and urban pupils tend to do better across all income deciles (Figure 5). Analysing GCSE performance data through the lens of SED shows how these ideas about rural and urban educational performance may be an artefact of SED rather than a consequence of rural living. When these factors are considered, there are persistent patterns of rural underperformance that have been (until recently [27–30]) overlooked.

On a like-for-like IDACI basis, rural pupils do less well in education outcomes across almost all IDACI deciles, though the gap is most extreme among the most disadvantaged (Figure 5). It is worth reinforcing this point.

As Howley [91] points out, a key conceptual failure lies in the construction of rurality as a mere 'setting' and this has enabled the rural–urban dichotomy to endure. Rurality is often presented as a binary construct [30,56,92,93] when it is not [27,47,62]. The binary rural–urban categories (RUC2011), as currently used, is not a sufficiently subtle system, as rurality is not simply determined by population. It is a construct based on more abstract social characteristics such as feelings of community and traditionalism [67], or more concrete features, such as landscapes or occupational structures [94,95] or access to school [96], transport, broadband [30], and post-16 options. Rural definitions that are based on a single demographic factor, such as population, lose sight of the diversity of rural and urban areas. The binary rural–urban RUC2011 definition deployed by the DfE does little to reflect this complexity and has obscured the patterns within the data. Rurality in education data should be considered across all the RUC2011 categories, and needs to recognise sparseness and other factors available in the full spread of the RUC2011 in Table 3. Adding sparseness (or remoteness or inaccessibility) within this analysis as proxies for reduced access to real choice-making opportunities [97–101] would enrich the analysis presented here. For example, in the national data, those rural pupils close to urban or cosmopolitan centres from more affluent backgrounds may have access to enhanced educational and employment options.

The prevailing narrative of English rural overperformance in average data [82] has hidden the underperformance of SED rural pupils from the political landscape. Presenting the 'average' performance of rural pupils as better than urban pupils (Table 4 and Figure 1) obscures the underlying performance patterns shown in Figure 5. Rural underachievement has been additionally masked by the relative sizes and distribution of rural and urban populations. When you take account of measures of SED, rural performance presents differently from the traditionally described narrative of urban underperformance. To develop strategies to address rural educational outcomes, the gap in attainment needs to be made more visible. While the rural gap was masked, SED rural pupils missed out on policy intervention as they were not seen [102]. This diverted attention away from the issue of rural educational outcomes and allowed a narrative that outcomes are better in rural settings to endure. This influenced educational policy and impacted national funding and resourcing decisions.

Moving beyond the explicit rural–urban dichotomy [103,104] that is present in the analysis of the DfE quantitative data in Table 4 to a more contextualised form through the lens of SED has generated a deeper insight into the patterns of rural attainment and educational progression [69,91]. But this is a small step using the data currently publicly available. Using measures of remoteness [105,106] or inaccessibility [26,107,108] would be useful in supporting schools in coastal and other remote communities [28,109]. In the short term, providing access to sparsity data within the DfE national database would quickly provide a more nuanced data analysis both nationally and within rural areas. Sparse, remote, or isolated rurality could be considered as a condition worthy of further investigation for the pupils in a wider rural area [110,111] to drive attention to this overlooked group

of rural pupils. While the DfE and DEFRA data have a limited capacity to illuminate the experiences of rural pupils due to the inelegant way they categorise urban and rural, they do allow for isolation of SED and rurality and educational performance.

The questions emerging from this analysis are deeper than identifying the patterns between rurality and educational outcome. We can identify these stubborn patterns in educational outcomes with the blunt measure of 'rural' and 'disadvantage' currently used in the DfE data, but this does little to lower the educational barriers for SED rural pupils. Noting the patterns in rural educational outcomes does not advance the knowledge as to how this can be converted to educational advantage for SED rural students. Though it has been recognised that it is far harder to use the available methodologies to improve participation effectively in rural areas [112], little has been done to situate the findings of rural education research in the communities where the research is gathered.

This research recognises several key contextual inconsistencies, for example, issues over the use of disadvantage are noted in the literature but the use of rural is used in such a way that it renders many of the international comparisons and even inter-department comparisons unworkable.

Secondly, and more significantly, as Howley [91] pointed out 20 years ago, the key conceptual failure lies in the construction of rural as a mere 'setting'. Rural and urban are not binary constructs. Rurality needs to be considered in both traditional DEFRA 'rural' terms (rural or urban by an agreed measure) and also recognise sparseness (or remoteness or inaccessibility) as a key part of some rural areas. These proxies for reduced access to real choice-making opportunities [97–101] and their relationship to educational outcomes need further exploration.

External factors limit progression, as illuminated in the work on gender inequality in the metaphorical glass ceiling [113], and the robust relationship between socioeconomic background and educational and social progress [114–120] gave rise to the notion of a class ceiling [34]. This analysis presents limiting conditions for rural pupils that illuminate the potential of a grass ceiling that exposes the limiting factors of place on rural pupils' experience of, and transition through, education.

While the DfE data are a blunt tool to illuminate the experiences of rural pupils, it is the only national data set available that allows for the isolation of SED and rurality, and within these data, we can see the emergence of a grass ceiling that appears to limit the educational outcomes for rural pupils. This inequality is particularly resonant as it has been seen through the de-contextualised lens so that a narrative of urban underperformance (rather than SED) has driven intervention policy to date [77,87,88].

We call for further analysis using the more nuanced measures of rurality and disadvantage to give greater attention to understand why rural pupils perform differently, and what the educational establishment should do differently to raise the grass ceiling and lower the impact of place-based disadvantage.

**Funding:** This research was funded by The ESRC Grant number [680047055].

**Institutional Review Board Statement:** The study was approved by the Ethics Committee of The University of Exeter 2019 STF/17/18/11.

**Informed Consent Statement:** Not applicable.

**Data Availability Statement:** Not applicable.

**Conflicts of Interest:** The author declares no conflict of interest.

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
