# Peer review of "The Grass Ceiling: Hidden Educational Barriers in Rural England"

_education, doi:10.3390/educsci14020165_

Round 1

Reviewer 1 Report

Comments and Suggestions for Authors

This is a thought-provoking paper but is written from a domestic viewpoint, with an international audience at a loss concerning the acronyms used. For example, RUC2011 should be defined earlier in the paper as the term ‘rural’ evokes different imagery for different people.

International readers will not be familiar with the IDACI. Definition required.

Request that the study authors review the paper through the eyes of an international audience as several assumptions are made concerning the English education system that an international audience, specifically an American audience, will not be familiar with, which impacts an understanding of the study.

Examples of the different policy definitions of “rurality” would be helpful. Issues relating to "nomenclature" are at the core of the paper. As reference is made in the paper to international research it would be relevant to discuss how other countries have handled this issue.

Query- What is the data source for the following statement? Around 82% of pupils taking GCSE exams in any year are urban, and 18% are 199 rural, with half of these rural pupils living in towns and fringes.

Graph presentation size etc, should be consistent.

Author Response

Thank you for the comprehensive feedback,  it has been invaluable in reshaping the submission and making it more internationally accessible. Amends made as noted in the text below in bold

This is a thought-provoking paper but is written from a domestic viewpoint, with an international audience at a loss concerning the acronyms used. For example, RUC2011 should be defined earlier in the paper as the term ‘rural’ evokes different imagery for different people.
Section added on defining rural in England 130-146

International readers will not be familiar with the IDACI. Definition required.
This has now been added to lines 84-92

Request that the study authors review the paper through the eyes of an international audience as several assumptions are made concerning the English education system that an international audience, specifically an American audience, will not be familiar with, which impacts an understanding of the study.

Additional clarity is now added with respect to the English system and use of internationally confusing terms (high school – public school, Higher Education, University etc) Other English centric terms not directly relevant to the study are removed (Progress 8, Attainment 8)

English High School and GCSEs are introduced and outlined in new lines 94-98

Examples of the different policy definitions of “rurality” would be helpful. Issues relating to "nomenclature" are at the core of the paper. As reference is made in the paper to international research it would be relevant to discuss how other countries have handled this issue.

Section added lines 131-178 on English Rurality

New section on international rurality measures added lines 209-233

 Query- What is the data source for the following statement? Around 82% of pupils taking GCSE exams in any year are urban, and 18% are 199 rural, with half of these rural pupils living in towns and fringes.

Source Added line 307

Graph presentation size etc, should be consistent.

All figures / tables now consistent with similar legends

Reviewer 2 Report

Comments and Suggestions for Authors

Comments to the authors

This manuscript focuses on secondary data analysis of data collected by the DfE around the factors that affect the discrepancy in academic attainment and outcomes between pupils who live in rural and urban areas of England.

Thank you for the opportunity to review this manuscript. I found this manuscript interesting to read and I think it has potential to contribute to the literature in several ways, especially around factors affecting educational attainment and outcomes. Abstract of this manuscript was very clear in outlining the key parts of this study.

Introduction

The literature review presented some initial details around the importance of educational achievement and the impact of socioeconomic disadvantage, as long-term factors affecting an individual’s life and opportunities. However, this section is extremely brief. I would strongly recommend further expanding on the key topics and areas of focus of this manuscript so that the reader is on the same page as the authors about what key information they need to be aware of. Some of the details (around the definition of disadvantage for example) at the beginning of the method section might fit better within the introduction instead.

It would be useful to highlight the research questions for this study clearly in this section, so that the reader has a clear idea of that before reading the rest of the manuscript.

Methods

This section presented some details around the databases used for this study, as well as some key methodological limitations.

I think this section still needs information around what exactly this study involved. I would recommend a detailed description of how data was collected from the databases, as well as any inclusion or exclusion criteria, as well as how data was compared. The manuscript at the moment assumes that the reader is an expert in comparative studies or secondary data analyses, which I think needs to be addressed. This would really support the accessibility and transparency of this manuscript.

In line 42, the manuscript is showing an error in showing a citation. This needs editing, if it is within the remit of the authors to edit this. The same error message also appears in line 83 and 137, so all of these require editing.

Results

The results have been presented somewhat clearly. In this section, the figures have not been presented in line with the text, as they seem to be positioned in non-suitable places with the text. Their respective legends also seem to be in other places, so this requires editing. This has impacted the clarity and presentation of the results in this section.

More details are also needed around the specifics of the data analysis that was conducted to reach these findings, as that is not clear at the moment. There are some details around interpretation of the results, but these are difficult to follow without the specifics of how data was analysed.

Conclusions

This section has provided some initial clear interpretations of the findings, as well as some links between the data, the results and relevant literature, but I think that further details are needed about the interpretation of the data and its links with literature. Further details around what these findings mean, as well as their implications maybe for practice or for further research would be highly beneficial. The key conclusions and their relevance to the wider literature need to be emphasised more.

Typos and other edits:

In line 42, the manuscript is showing an error in showing a citation. The same error message also appears in line 83 and 137, as well as in many other places later in the manuscript. This needs editing, if it is within the remit of the authors to edit this.

I also think there are several significant errors in the formatting of the reference list, as the formatting of the details in each reference is not consistent throughout and some references seem to be missing some details.

Author Response

Thank you for the comprehensive feedback, it has been invaluable in reshaping the submission. Amends made as noted in the text below in bold

This manuscript focuses on secondary data analysis of data collected by the DfE around the factors that affect the discrepancy in academic attainment and outcomes between pupils who live in rural and urban areas of England.Thank you for the opportunity to review this manuscript. I found this manuscript interesting to read and I think it has potential to contribute to the literature in several ways, especially around factors affecting educational attainment and outcomes. Abstract of this manuscript was very clear in outlining the key parts of this study.

Introduction
The literature review presented some initial details around the importance of educational achievement and the impact of socioeconomic disadvantage, as long-term factors affecting an individual’s life and opportunities. However, this section is extremely brief. I would strongly recommend further expanding on the key topics and areas of focus of this manuscript so that the reader is on the same page as the authors about what key information they need to be aware of. Some of the details (around the definition of disadvantage for example) at the beginning of the method section might fit better within the introduction instead.
Section moved to introduction and added context on outcomes section added lines 37-76

It would be useful to highlight the research questions for this study clearly in this section, so that the reader has a clear idea of that before reading the rest of the manuscript.
RQ added at the end of the introduction

Methods
This section presented some details around the databases used for this study, as well as some key methodological limitations.

I think this section still needs information around what exactly this study involved. I would recommend a detailed description of how data was collected from the databases, as well as any inclusion or exclusion criteria, as well as how data was compared. The manuscript at the moment assumes that the reader is an expert in comparative studies or secondary data analyses, which I think needs to be addressed. This would really support the accessibility and transparency of this manuscript.
Sections added on the key variables and data extraction lines 84-92
Additional clarity is now added with respect to the English system and use of internationally confusing terms (high school – public school, Higher Education, University etc) Other English centric terms not directly relevant to the study are removed (Progress 8, Attainment 8)

English High School and GCSEs are introduced and outlined in new lines 94-98. Section added lines 131-178 on English Rurality
New section on international rurality measures added lines 209-233

In line 42, the manuscript is showing an error in showing a citation. This needs editing, if it is within the remit of the authors to edit this. The same error message also appears in line 83 and 137, so all of these require editing.
These table and figure references have been replaced

Results
The results have been presented somewhat clearly. In this section, the figures have not been presented in line with the text, as they seem to be positioned in non-suitable places with the text. Their respective legends also seem to be in other places, so this requires editing. This has impacted the clarity and presentation of the results in this section.
Figures have been formatted back to the submitted version layout, with consistent size and text

More details are also needed around the specifics of the data analysis that was conducted to reach these findings, as that is not clear at the moment. There are some details around interpretation of the results, but these are difficult to follow without the specifics of how data was analysed.
Lines 88-95 added to include this additional data.

Conclusions
This section has provided some initial clear interpretations of the findings, as well as some links between the data, the results and relevant literature, but I think that further details are needed about the interpretation of the data and its links with literature. Further details around what these findings mean, as well as their implications maybe for practice or for further research would be highly beneficial. The key conclusions and their relevance to the wider literature need to be emphasised more.
New lines added in conclusions to emphasise the main findings. 376-393 and 409-459 re-edited accordingly with new material added.

Typos and other edits:
In line 42, the manuscript is showing an error in showing a citation. The same error message also appears in line 83 and 137, as well as in many other places later in the manuscript. This needs editing, if it is within the remit of the authors to edit this. 
There was an issue with the copy sent to the reviewers and the tables and figures did not display as anticipated. Also the referencing of all the figures and tables was removed, and this led to errors in the formatting. This has been addressed.

I also think there are several significant errors in the formatting of the reference list, as the formatting of the details in each reference is not consistent throughout and some references seem to be missing some details.I have revied each reference and made amends here

Round 2

Reviewer 2 Report

Comments and Suggestions for Authors

Comments to the authors

I would like to thank the authors for their hard work in revising this manuscript. It is on a very interesting and important field, so I am glad that they have continued working on publishing their findings. I really appreciate the detailed responses to my comments on the first version and I am happy with all the changes that have been made. I am looking forward to reading the final version when it is published.